# Sixteen-year trends in multiple lifestyle risk behaviours by socioeconomic status from 2004 to 2019 in New South Wales, Australia

**Binh Nguyen** [1,2]*, **Janette Smith**[1,3], **Philip Clare**[1,2,4], **Philayrath Phongsavan**[1,2], **Leonie Cranney** [1,2¤], **Ding Ding**[1,2]

**1** Prevention Research Collaboration, Sydney School of Public Health, Faculty of Medicine and Health, The University of Sydney, Camperdown, NSW, Australia, **2** Charles Perkins Centre, Charles Perkins Centre, The University of Sydney, Camperdown, NSW, Australia, **3** NSW Biostatistics Training Program, NSW Ministry of Health, St Leonards, NSW, Australia, **4** National Drug and Alcohol Research Centre, UNSW Sydney, Sydney, NSW, Australia

¤ Current address: School of Public Health, Faculty of Health, University of Technology Sydney, Sydney, NSW, Australia

* thanh-binh.nguyen-duy@sydney.edu.au

**Data Availability Statement:** All data reported in this study have been included in the manuscript or as supplementary information.

## Abstract

Few studies have examined trends in inequalities related to lifestyle risk behaviours. This study examined 1) 16-year (2004–2019) trends of individual lifestyle risk factors and a combined lifestyle risk index and 2) trends in socioeconomic inequalities in these risk factors, in New South Wales (NSW; Australia) adults. Data was sourced from the NSW Adult Population Health Survey, an annual telephone survey of NSW residents aged ≥16 years, totalling 191,905 completed surveys. Excessive alcohol consumption, current smoking, insufficient physical activity, insufficient fruit and/or vegetable consumption, sugar-sweetened beverage [SSB] consumption, and a combined lifestyle risk index (overall high-risk lifestyle defined as total number of lifestyle risk behaviours ≥2) were examined. Socioeconomic status was assessed using education attainment, postal area-level disadvantage measured by Index of Relative Socioeconomic Disadvantage (IRSD), and remoteness based on Accessibility-Remoteness Index of Australia Plus (ARIA+). Socioeconomic inequalities were examined as prevalence difference for absolute inequalities and prevalence ratio for relative inequalities. The prevalence of lifestyle behaviours by levels of each socioeconomic status variable were estimated using predicted probabilities from logistic regression models. After adjusting for covariates, there was a decrease in prevalence over time for most lifestyle risk behaviours. Between 2004 and 2019, the prevalence decreased for current smoking from 21.8% to 17.1%, insufficient physical activity from 39.1% to 30.9%, excessive alcohol consumption from 15.4% to 13.7%, daily SSB consumption from 29.9% to 21.2%, and overall high-risk lifestyle from 50.4% to 43.7%. Socioeconomic inequalities, based on one or more of the socioeconomic variables, increased over time for current smoking, insufficient physical activity, daily SSB consumption, and an overall high-risk lifestyle. Overall, the health behaviours of the NSW population improved between 2004 and 2019. However, some socioeconomic inequalities increased during this time, highlighting the need for effective public health

**Funding:** The authors received no specific funding for this work.

**Competing interests:** The authors have declared that no competing interests exist.

strategies that seek to improve health behaviours among the most socioeconomically disadvantaged.

## Introduction

Non-communicable diseases are the leading causes of death worldwide, accounting for 71% of deaths globally [1]. A substantial proportion of disease burden is attributed to modifiable risk factors such as alcohol consumption, tobacco use, an unhealthy diet and physical inactivity, which individually and jointly affect health and longevity [2–9]. Reducing lifestyle risk is critical to preventing premature morbidity and mortality as well as lowering healthcare costs [1].

Understanding how lifestyle risk behaviours, individually and jointly, change over time in the population can help track long-term health trends, identify problem areas, benchmark progress, and evaluate historic and existing initiatives, in order to inform decision making [10, 11]. In addition, monitoring trends in lifestyle risk factors across the social gradient could track inequalities, identify groups in need and inform equitable resource allocation. In fact, widening inequalities could be an unintended consequence of public health interventions as those who need interventions the most tend to have the least opportunities, resources and capacity to benefit from interventions, as hypothesised by the 'inverse care law' [12, 13]. Empirical evidence from several Western countries suggests that despite overall improvement at a population level, inequalities are increasing for various lifestyle behaviours [10, 11, 14–16]. Social inequalities are also widening for health outcomes including premature mortality and self-reported health [17], and countries worldwide have considered reducing social disparities a priority [18, 19].

There has been limited recent evidence on how lifestyle behaviours have collectively changed over time in various countries in the general population and in different socioeconomic subgroups. Two previous studies from Australia [10, 14] and three studies from European countries [11, 15, 16] have examined trends in multiple lifestyle behaviours more than a decade ago. A previous Australian study examining trends in the New South Wales (NSW) adult population found a widening socioeconomic gap in health risk behaviours for the period 2002 to 2012, despite the apparent overall improvement in some lifestyle risk factors [10]. Since 2012, there has been a number of public health policies, strategies and successive preventive programs implemented in NSW to address modifiable risk factors. It is vital to track whether these population-wide efforts have had an impact on lifestyle behaviours, measurable at a population level. Using population representative data from NSW, Australia, this study aims to: 1) examine the long-term trends of individual lifestyle risk factors (excessive alcohol consumption, current smoking, insufficient physical activity, insufficient fruit and/or vegetable consumption, sugar-sweetened beverage [SSB] consumption) and combined lifestyle risk index from 2004 to 2019, and 2) investigate whether trends in these risk factors vary by socioeconomic inequality, based on education attainment, area-level disadvantage and remoteness.

## Methods

### Ethics statement

The NSW Population Health survey was approved by the NSW Population and Health Services Research Ethics Committee. Participants provided informed verbal consent.

## Data source and survey population

The NSW Population Health Survey is an annual survey of NSW residents, conducted using a computer-assisted telephone interview system by trained interviewers [20, 21]. The survey includes questions regarding the health behaviours, health status and outcomes of NSW adults aged 16 years and over. Households from each health administrative area are contacted using list-assisted random digit dialling and one eligible resident from the selected household is randomly selected and invited to participate in the survey. A landline sampling frame was used prior to 2012, with mobile phone numbers added to landline phone numbers in an overlapping dual-frame design from 2012 onwards [20]. The addition of mobile phone numbers did not have a negative impact on response rates and data collection, and the demographic profile of respondents from the two frames combined was similar to the NSW population profile [20].

## Measures

*Socioeconomic status*. Socioeconomic status was assessed using three measures, including both an individual-level indicator (education attainment) and two area-level indicators (socioeconomic disadvantage and geographic remoteness), designed to capture different aspects of socioeconomic status. *Education attainment* was the highest education level completed and categorised as: school certificate (10 years) or lower, high school/trade/diploma (12 years), and university degree or higher. *Area-level disadvantage* was coded from respondents' postcodes using the Socio-Economic Indexes for Areas (SEIFA) Index of Relative Socioeconomic Disadvantage (IRSD), a commonly used indicator of socioeconomic status in Australia [22]. This composite measure includes area-based population attributes such as high unemployment, low income, low education attainment, and jobs in relatively unskilled occupations. The level of disadvantage was categorised as quintiles, based on state-level distribution and the 2016 SEIFA IRSD data [23] (assigned from postcode) with quintile 1 being the least disadvantaged and quintile 5 the most disadvantaged; the middle three quintiles were grouped together for our analysis. *Geographical remoteness*, which has been linked with social disadvantages [24], was measured using the postcode-level Accessibility-Remoteness Index of Australia Plus (ARIA+) [25], which reflect the ease or difficulty with which one can access a range of services, with values ranging from 0 (high accessibility) to 15 (high remoteness). The index was categorised as major city vs. regional/remote.

*Lifestyle risk behaviours*. Five lifestyle risk behaviours were examined: alcohol consumption, current smoking, insufficient physical activity, insufficient fruit and/or vegetable consumption, and SSB consumption. The NSW PHS survey questions about these risk behaviours are presented in S1 File. The questions about physical activity were based on the Active Australia Survey, which was found to have acceptable reliability and validity [26]. All variables were dichotomised as 'at risk' and 'not at risk'. Specifically, alcohol risk was defined as consuming more than 10 drinks per week, based on the 2020 alcohol guidelines for the general population [27]. Smoking risk was based on being a current tobacco smoker, or current user of e-cigarettes, which included self-reported daily and occasional use. Physical activity risk (insufficient physical activity) was defined as not meeting the minimal recommendation of 150 minutes of moderate-to-vigorous intensity physical activity per week (with vigorous physical activity weighted by two) [28]. Insufficient fruit and/or vegetable consumption was used as a marker for dietary risk, defined as consuming less than two servings of fruit or less than three servings of vegetables a day. Although the Australian Dietary Guidelines recommend five servings of vegetables per day, previous research based on NSW data has examined consumption trends using three serves per day due to the low percentage of the population meeting this recommendation [29]. At-risk SSB consumption was defined as consuming one SSB per day or more.

Finally, a dichotomous lifestyle risk index was developed based on the total number of individual risk behaviours for smoking, alcohol, physical activity and fruit and vegetable intake. A high lifestyle risk was defined as engaging in two or more lifestyle risk behaviours, as used in previous studies [10, 29]. Daily SSB consumption was included in the lifestyle risk index in supplementary analyses only, as the question relating to this variable was not asked in 2004, 2005, 2011, and 2013.

## Covariates

To account for demographic changes in the NSW adult population over time, we adjusted for the following variables when modelling the relationship between time and lifestyle risk indicators: age, sex, country of birth (categorised into three groups based on the very high/high/medium levels of human development index of the country [30]), whether they spoke a language other than English at home, and employment status. These covariates were selected because the prevalence of multiple risk factors has been shown to vary based on different sociodemographic characteristics such as sex, age groups, employment status, immigration status and cultural and linguistic backgrounds. Country of birth and whether a language other than English is spoken at home are routinely used by governments in Australia as indicators for cultural and linguistic diversity.

## Statistical analysis

We used 16 years of data from 2004 to 2019. For descriptive analyses, we present unweighted sample characteristics. For the remaining analysis, we estimated prevalence of individual risk behaviours and the summary lifestyle risk indicator for each year, using the marginal predicted probabilities from a logistic regression model of each risk behaviour. Each year's data was weighted using post-stratification weights calculated by the NSW Ministry of Health (see https://www.health.nsw.gov.au/surveys/adult/Pages/default.aspx) to be population representative and to ensure standard error estimates account for the complex sample design. We then estimated prevalence of individual risk behaviours and the summary lifestyle risk indicator by levels of education attainment, area-level disadvantage, and remoteness using the same method, but including a time by socioeconomic status interaction term to allow prevalence to vary by levels of each socioeconomic status variable and time. In addition, we present socioeconomic inequalities based on the estimated prevalence, as absolute inequality (prevalence difference) and relative inequality (prevalence ratio), both meaningful measures to monitor inequalities [31, 32]. For all analyses, we a-priori set the α at 0.05, and we present 95% confidence limits.

Supplementary analyses included daily SSB consumption as an additional lifestyle risk behaviour examined as part of the summary lifestyle risk indicator. In addition, the prevalence of being a current tobacco smoker and the prevalence of being a current user of e-cigarettes were estimated separately for each year. Any missing data, such as those resulting from the split survey design trialled in 2007 where respondents were randomly allocated a different set of questions to enhance completion rates, as well as questions where the respondent declined to answer, were handled using multiple imputation with the mice package in R [33]. To reduce potential bias, all analyses were conducted using multiple imputation using fully conditional specification, with M = 40 imputations [34], with imputations that accounted for complex survey design (sampling weights and strata) [35]. Further details on the multiple imputation methods used can be found in S2 File and S1 Table. All analyses were conducted in Stata 16.1 [36] and R 4.03 [37].

## Results

### Sample characteristics

From 2004 to 2019, 191,905 surveys were completed with the number of survey respondents varying between 7,962 and 13,521 per year (Table 1). Across survey years, the mean age of the sample ranged from 52 (SD: 18) to 61 (SD: 18) years, and the proportion of males varied between 37.0% and 43.8% in the unweighted descriptive analysis. Overall, there was an increase in respondents' educational level with the proportion of participants with a school certificate decreasing from 31.1% in 2004 to 24.3% in 2019, and the proportion of those with a university degree increasing from 22.4% to 30.3% in the same period. Around 11–15% of the sample was from postal areas with the least disadvantaged quintile and approximately 20–26% fell within the most disadvantaged quintile. The majority of the sample (49–60%) resided in major cities, around a third (29–34%) resided in inner regional areas, and more than a tenth (11–17%) were from outer regional areas.

### Trends in individual risk factors and combined lifestyle risk index

Except for insufficient fruit and vegetable consumption, the prevalence of all individual risk factors showed a decreasing trend between 2004 and 2019 (Fig 1, S2 Table). After adjusting for covariates, the following decreases in prevalence between 2004 and 2019 were substantial: current smoking from 21.8% (95% confidence intervals [CI]: 20.7–22.9) to 17.1% (95% CI: 15.9–18.4), insufficient physical activity from 39.1% (95% CI: 37.8–40.4) to 30.9% (95% CI: 29.5–32.3), and daily SSB consumption from 29.9% (95% CI: 28.6–31.2) in 2006 (first year of data for SSB) to 21.2% (95% CI: 19.9–22.5) in 2019. The prevalence for excessive alcohol consumption slightly decreased from 15.4% (95% CI: 14.5–16.4) in 2014 to 13.7% (95% CI: 12.7–14.7) in 2019. The prevalence for insufficient fruit and vegetable consumption fluctuated between 2004 and 2019, and ranged from 70.1% (95% CI: 69.8–72.0) to 81.7% (95% CI: 80.6–82.9). The prevalence of high overall lifestyle risk, based on having two or more individual lifestyle risk factors, decreased from 50.4% (95% CI: 49.0–51.8) in 2004 to 43.7% (95% CI: 42.2–45.3) in 2019. Supplementary analyses showed that the decrease in prevalence of current smoking primarily reflected a decrease in current tobacco smoking (S1 Fig).

### Trends in socioeconomic inequalities in lifestyle risk factors

**Individual lifestyle risk factors.** The prevalence, prevalence differences and ratios (and their 95% CI) by education attainment, area-level disadvantage and remoteness are shown in Figs 2 to 4 and S3–S5 Tables.

For current smoking, the prevalence decreased for the two least disadvantaged education subgroups over time. There was a suggestion of a decrease for the most disadvantaged subgroup, although it was not significant. Throughout the study period, the prevalence remained higher for the most disadvantaged subgroup compared to the least disadvantaged ones in terms of education and area-level disadvantage. The difference was less substantial in terms of remoteness. Between 2004 and 2019, there was no substantial inequality in smoking prevalence by remoteness, a relatively stable inequality by education, but a slight trend for increasing absolute and relative inequalities by area-level disadvantage (for the middle disadvantage subgroup, a prevalence difference of 4.3 percentage points [95% CI: 1.3–7.4] in 2004 to 8.0 percentage points [95% CI: 5.5–10.5] in 2019, and a prevalence ratio of 1.2 [95% CI: 1.0–1.4] in 2004 to 1.9 [95% CI: 1.4–2.3] in 2019; for the most disadvantaged subgroup, a prevalence difference of 7.5 percentage points [95% CI: 3.9–11.2] in 2004 to 16.8 percentage points [95% CI:

**Table 1. Sociodemographic and lifestyle characteristics for adults (aged 16 years and over) from the NSW Adult Population Health Survey, 2004–2019.**

| Variable | Categories | Years | | | | | | | | | | | | | | | |
|---|---|---|---|---|---|---|---|---|---|---|---|---|---|---|---|---|---|
| | | 2004 | 2005 | 2006 | 2007[a] | 2008 | 2009 | 2010 | 2011 | 2012 | 2013 | 2014 | 2015 | 2016 | 2017 | 2018 | 2019 |
| Age (mean years; SD) | | 52 (18) | 53 (18) | 53 (18) | 54 (18) | 55 (18) | 55 (18) | 56 (17) | 57 (17) | 54 (18) | 54 (18) | 56 (18) | 55 (19) | 59 (18) | 59 (18) | 60 (18) | 61 (18) |
| Males (%) | | 41.0 | 40.0 | 40.7 | 38.8 | 40.0 | 38.2 | 37.7 | 37.0 | 41.1 | 41.6 | 41.5 | 43.4 | 41.3 | 43.5 | 43.0 | 43.8 |
| Educational attainment (%) | School certificate | 31.1 | 30.1 | 28.0 | 36.8 | 36.8 | 35.8 | 35.0 | 35.1 | 30.0 | 27.0 | 28.5 | 28.3 | 30.1 | 27.0 | 25.9 | 24.3 |
| | High school/ trade/diploma | 44.9 | 44.7 | 45.6 | 36.2 | 37.8 | 37.2 | 37.6 | 37.1 | 39.4 | 40.5 | 38.0 | 38.1 | 37.4 | 41.5 | 41.8 | 44.0 |
| | University or higher | 22.4 | 23.6 | 24.6 | 25.4 | 24.1 | 25.7 | 26.3 | 26.7 | 29.4 | 31.2 | 32.1 | 31.9 | 31.5 | 30.0 | 31.0 | 30.3 |
| Index of relative socio-economic disadvantage (IRSD) (%) | Quintile 1 (least disadvantaged) | 11.5 | 14.4 | 14.1 | 14.4 | 14.8 | 14.6 | 14.5 | 14.0 | 13.4 | 15.1 | 13.9 | 13.4 | 12.8 | 13.3 | 13.7 | 14.2 |
| | Quintile 2 | 15.1 | 16.1 | 17.2 | 17.1 | 16.6 | 16.8 | 16.8 | 17.9 | 17.4 | 17.7 | 17.9 | 17.3 | 16.8 | 17.2 | 16.8 | 17.5 |
| | Quintile 3 | 18.7 | 19.5 | 18.9 | 19.8 | 20.1 | 20.3 | 19.1 | 18.5 | 18.7 | 19.3 | 19.1 | 18.4 | 19.4 | 19.5 | 19.9 | 19.0 |
| | Quintile 4 | 28.8 | 28.6 | 27.1 | 27.2 | 28.4 | 28.0 | 28.9 | 25.5 | 27.4 | 27.3 | 25.5 | 27.4 | 26.7 | 26.2 | 26.3 | 27.3 |
| | Quintile 5 (most disadvantaged) | 25.8 | 21.4 | 21.3 | 21.4 | 20.1 | 20.3 | 20.6 | 23.9 | 22.6 | 20.7 | 23.5 | 23.1 | 24.0 | 23.8 | 23.2 | 22.0 |
| Accessibility-Remoteness Index of Australia Plus (ARIA+) (%) | Major cities | 48.5 | 52.8 | 54.2 | 56.0 | 55.9 | 55.5 | 55.1 | 54.7 | 56.2 | 59.8 | 56.8 | 54.7 | 55.3 | 55.3 | 55.0 | 54.5 |
| | Inner regional | 32.1 | 34.0 | 31.5 | 31.9 | 32.6 | 32.6 | 32.9 | 29.7 | 28.5 | 27.6 | 29.4 | 29.7 | 29.7 | 29.8 | 29.8 | 30.5 |
| | Outer regional | 16.5 | 12.1 | 11.6 | 11.2 | 10.6 | 10.8 | 11.0 | 14.6 | 13.6 | 11.3 | 13.1 | 14.5 | 13.9 | 14.1 | 14.4 | 14.3 |
| | Remote/Very remote | 2.8 | 1.1 | 1.2 | 0.9 | 0.9 | 1.0 | 0.9 | 0.9 | 1.1 | 1.3 | 0.7 | 0.7 | 0.8 | 0.8 | 0.7 | 0.7 |
| Excessive alcohol consumption[b] (%) | No | 83.5 | 84.8 | 83.6 | 48.0 | 69.1 | 83.4 | 84.8 | 84.8 | 86.2 | 86.8 | 86.1 | 85.7 | 86.1 | 84.2 | 84.1 | 83.5 |
| | Yes | 16.0 | 14.6 | 15.9 | 8.2 | 13.3 | 16.1 | 14.6 | 14.7 | 13.3 | 12.4 | 13.2 | 13.4 | 13.4 | 14.9 | 14.8 | 15.0 |
| Insufficient physical activity[c] (%) | No | 58.7 | 57.1 | 57.3 | 23.4 | 46.8 | 56.7 | 54.8 | 55.9 | 4.2 | 54.7 | 56.6 | 56.8 | 54.8 | 53.3 | 54.0 | 51.2 |
| | Yes | 41.3 | 42.1 | 37.8 | 15.4 | 31.0 | 36.5 | 37.6 | 39.2 | 14.8 | 38.5 | 36.2 | 36.9 | 38.9 | 36.3 | 35.8 | 35.1 |
| Insufficient fruit and/or vegetable consumption[d] (%) | No | 24.1 | 28.9 | 30.9 | 17.8 | 27.0 | 33.6 | 32.3 | 29.3 | 28.1 | 27.7 | 27.8 | 24.0 | 25.1 | 22.5 | 20.7 | 20.0 |
| | Yes | 75.3 | 70.5 | 68.0 | 37.8 | 55.0 | 64.6 | 65.5 | 69.1 | 69.7 | 70.1 | 69.7 | 74.9 | 74.3 | 75.7 | 77.8 | 77.7 |
| Current smoking[e] (%) | No | 79.3 | 81.6 | 83.3 | 47.4 | 71.4 | 84.8 | 85.3 | 85.9 | 84.8 | 85.8 | 86.0 | 86.6 | 87.5 | 87.6 | 88.1 | 87.6 |
| | Yes | 20.6 | 18.3 | 16.6 | 9.6 | 13.7 | 15.0 | 14.6 | 13.9 | 15.0 | 14.0 | 13.8 | 13.0 | 12.4 | 12.0 | 11.6 | 11.9 |
| Daily sugar-sweetened beverage consumption[f] (%) | No | N/A | N/A | 74.0 | 42.3 | 61.8 | 55.5 | 56.2 | N/A | 78.7 | N/A | 80.5 | 80.4 | 82.6 | 81.7 | 83.3 | 83.3 |
| | Yes | N/A | N/A | 24.8 | 13.4 | 19.1 | 19.0 | 16.2 | N/A | 20.7 | N/A | 19.0 | 18.1 | 16.7 | 17.5 | 16.0 | 15.4 |
| High total lifestyle risk[g] | No | 49.5 | 52.1 | 52.2 | 3.3 | 23.8 | 52.8 | 52.0 | 51.7 | 6.3 | 51.1 | 51.8 | 51.3 | 50.8 | 47.6 | 46.9 | 44.2 |
| | Yes | 49.3 | 45.9 | 41.5 | 2.3 | 17.7 | 38.5 | 38.5 | 41.8 | 11.9 | 39.9 | 38.7 | 40.9 | 41.9 | 39.9 | 40.9 | 39.5 |
| Completeness (%) | | 97.3 | 96.3 | 90.1 | 5.5 | 39.9 | 67.5 | 65.0 | 92.5 | 17.8 | 90.1 | 89.2 | 89.8 | 91.2 | 86.1 | 86.6 | 82.3 |

*(Continued)*

**Table 1.** (Continued)

| | | | | | | | | | | | | | | | | |
|---|---|---|---|---|---|---|---|---|---|---|---|---|---|---|---|---|
| Total respondents (n) | | 9,786 | 11,500 | 7,962 | 13,178 | 10,296 | 10,719 | 10,245 | 13,041 | 13,269 | 13,027 | 12,687 | 13,393 | 13,521 | 13,300 | 13,177 | 12,804 |

Abbreviations: SD, standard deviation.

[a] For 2007, a survey split questionnaire design was used.

[b] Defined as more than 10 drinks per week.

[c] Defined as less than 150 minutes of moderate-vigorous physical activity per week.

[d] Defined as less than 2 servings of fruit per week and/or less than 3 servings of vegetables per week.

[e] Based on being a current tobacco smoker and/or current e-cigarette user.

[f] Based on consuming one sugar-sweetened beverage per day or more.

[g] Defined as engaging in two or more lifestyle risk behaviours, based on the following four individual risk behaviours: excessive alcohol consumption, insufficient physical activity, insufficient fruit and/or vegetable consumption, and current smoking.

12.7–20.8] in 2019, and a prevalence ratio of 1.4 [95% CI: 1.2–1.7] in 2004 to 2.8 [95% CI: 2.1–3.5] in 2019).

For excessive alcohol consumption, the decrease in prevalence over time was not different between socioeconomic groups based on education. Counterintuitively, for area-level disadvantage, there was a tendency for the least disadvantaged subgroup to have a higher prevalence of excessive alcohol consumption than the two most disadvantaged subgroups and this difference remained steady over time (relative to the least disadvantaged subgroup, the prevalence difference for the middle disadvantaged subgroup was -3.4 percentage points [95% CI: -6.2 to

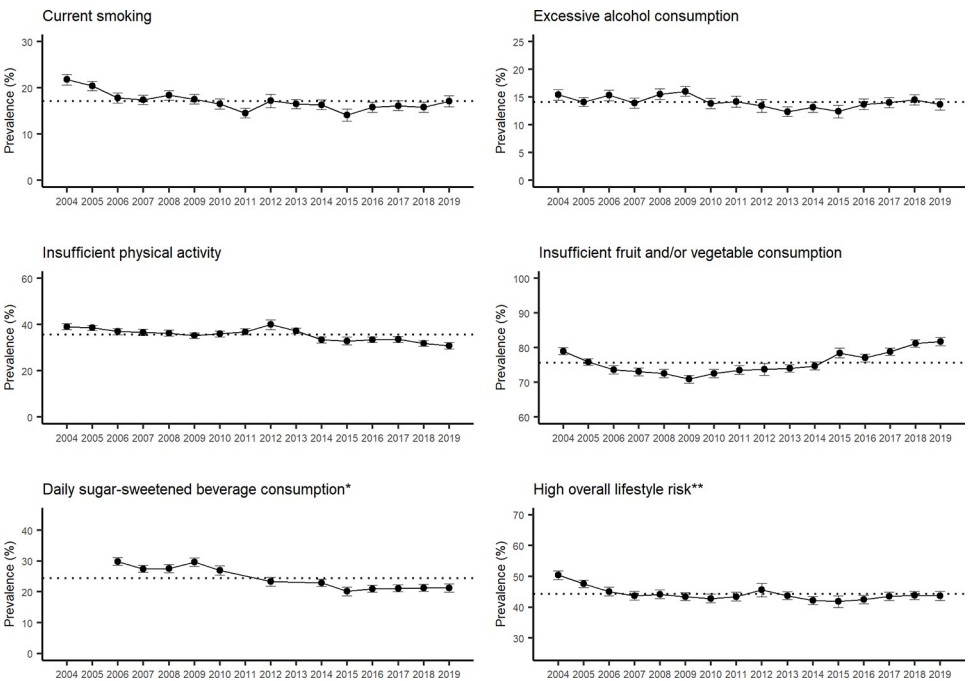

**Fig 1. Prevalence of individual and combined lifestyle risk factors, with 95% confidence intervals, by year in NSW adults aged 16 years and over, 2004–2019.** The dotted horizontal line represents the mean prevalence over the years 2014–2019. *The survey question related to daily sugar-sweetened beverage consumption was not asked in 2004, 2005, 2011 and 2013. ** Based on excessive alcohol consumption, insufficient physical activity, insufficient fruit and/or vegetable consumption and current smoking.

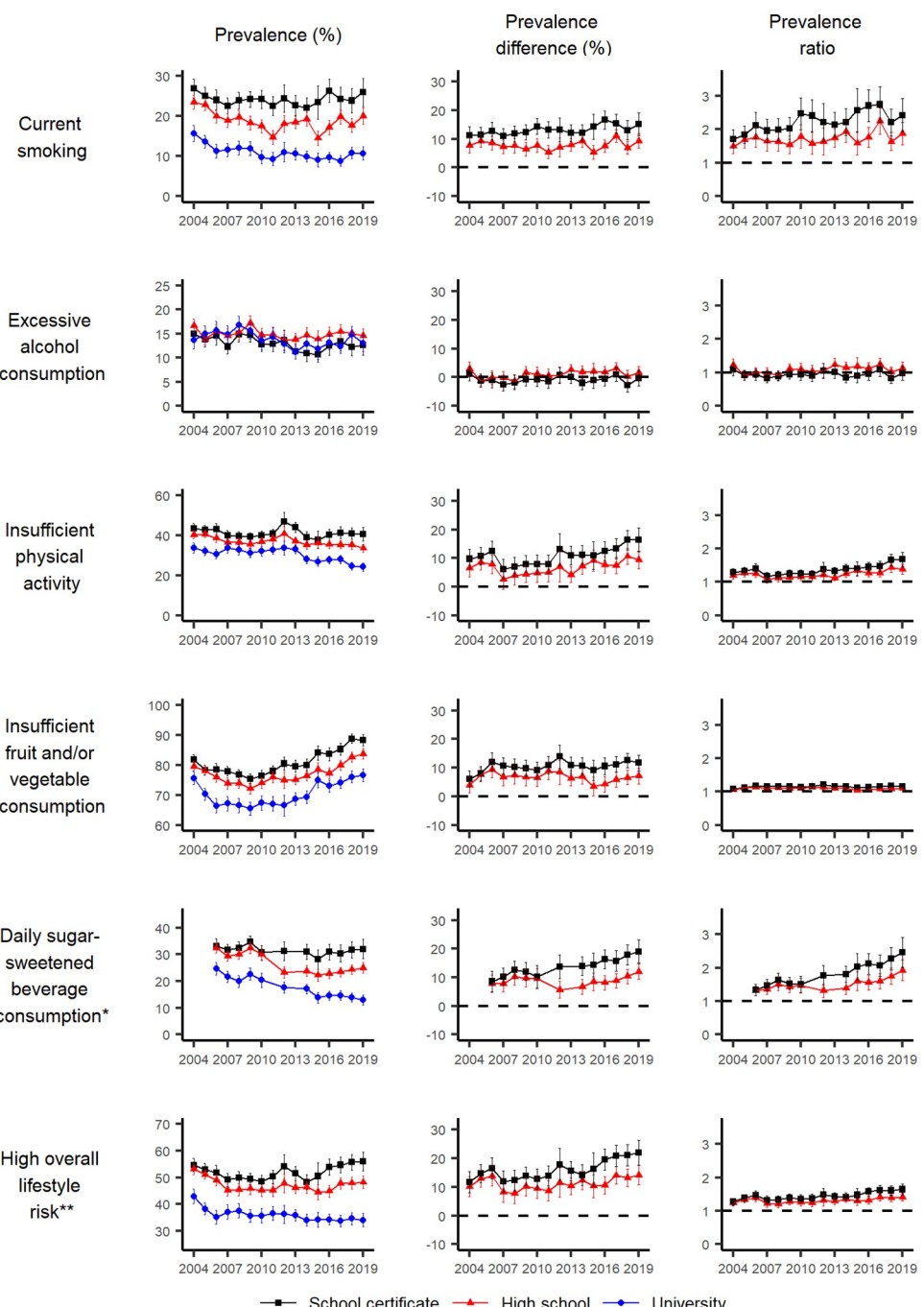

**Fig 2. Prevalence, prevalence difference and prevalence ratio (relative to university), with 95% confidence intervals, of individual lifestyle behaviours and combined lifestyle risk index for the most and least disadvantaged groups for education, by year, NSW adults aged 16 years and over, 2004–2019.** *The survey question related to daily sugar-sweetened beverage consumption was not asked in 2004, 2005, 2011 and 2013. **Based on current smoking, excessive alcohol consumption, insufficient physical activity, and insufficient fruit and/or vegetable consumption.

-0.6] in 2004 and -3.7 percentage points [95% CI: -6.4 to -1.0] in 2019, and the prevalence difference for the most disadvantaged subgroup was -4.7 percentage points [95% CI: -7.8 to -1.6] in 2004 and -4.0 percentage points [95% CI: -7.4 to -0.7] in 2019). Between 2004 and 2019,

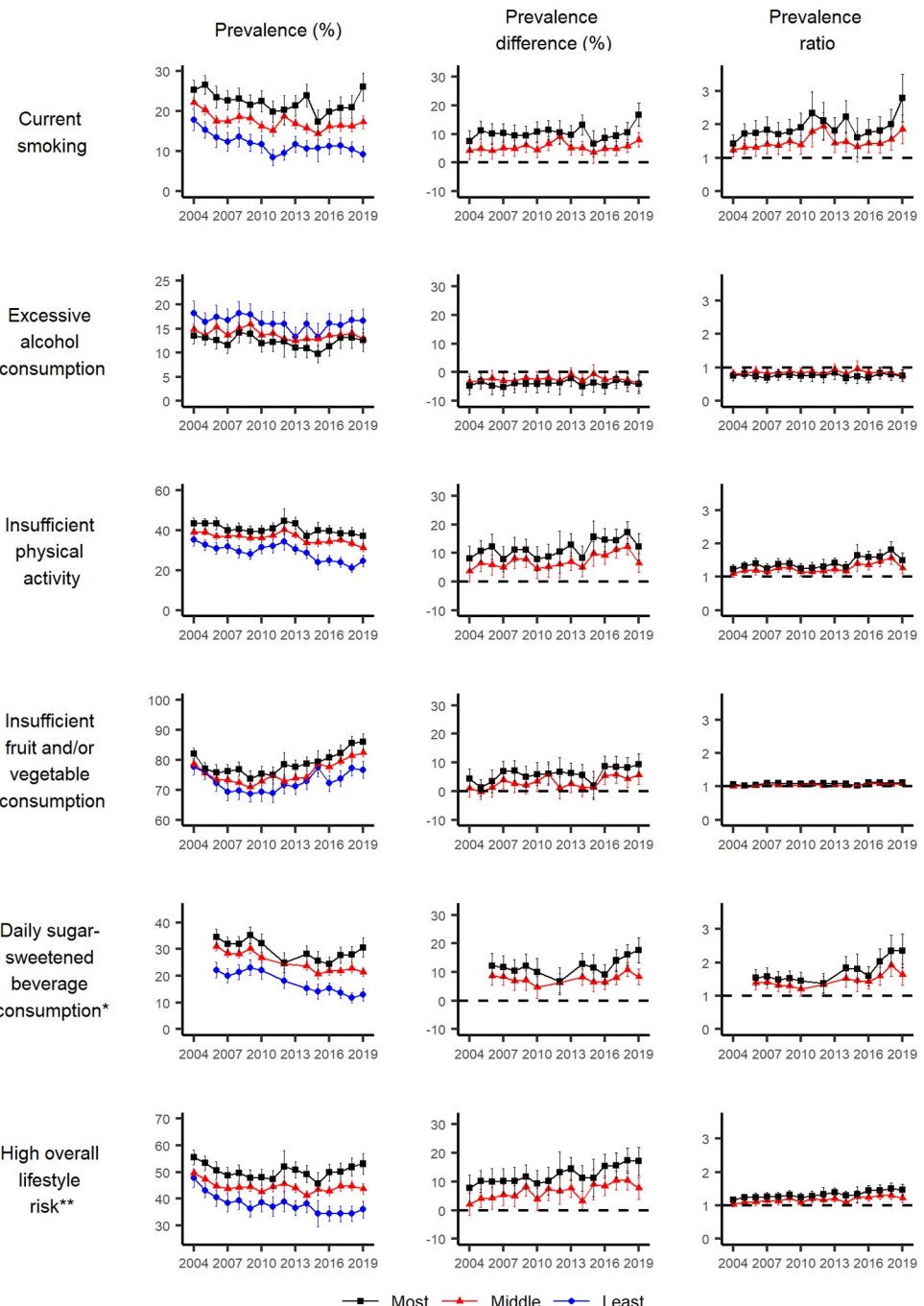

**Fig 3. Prevalence, prevalence difference and prevalence ratio (relative to the least disadvantaged group), with 95% confidence intervals, of individual lifestyle behaviours and combined lifestyle risk index for the most and least disadvantaged groups for area-level disadvantage (IRSD), by year, NSW adults aged 16 years and over, 2004–2019.** *The survey question related to daily sugar-sweetened beverage consumption was not asked in 2004, 2005, 2011 and 2013. ** Based on excessive alcohol consumption, insufficient physical activity, insufficient fruit and/or vegetable consumption and current smoking.

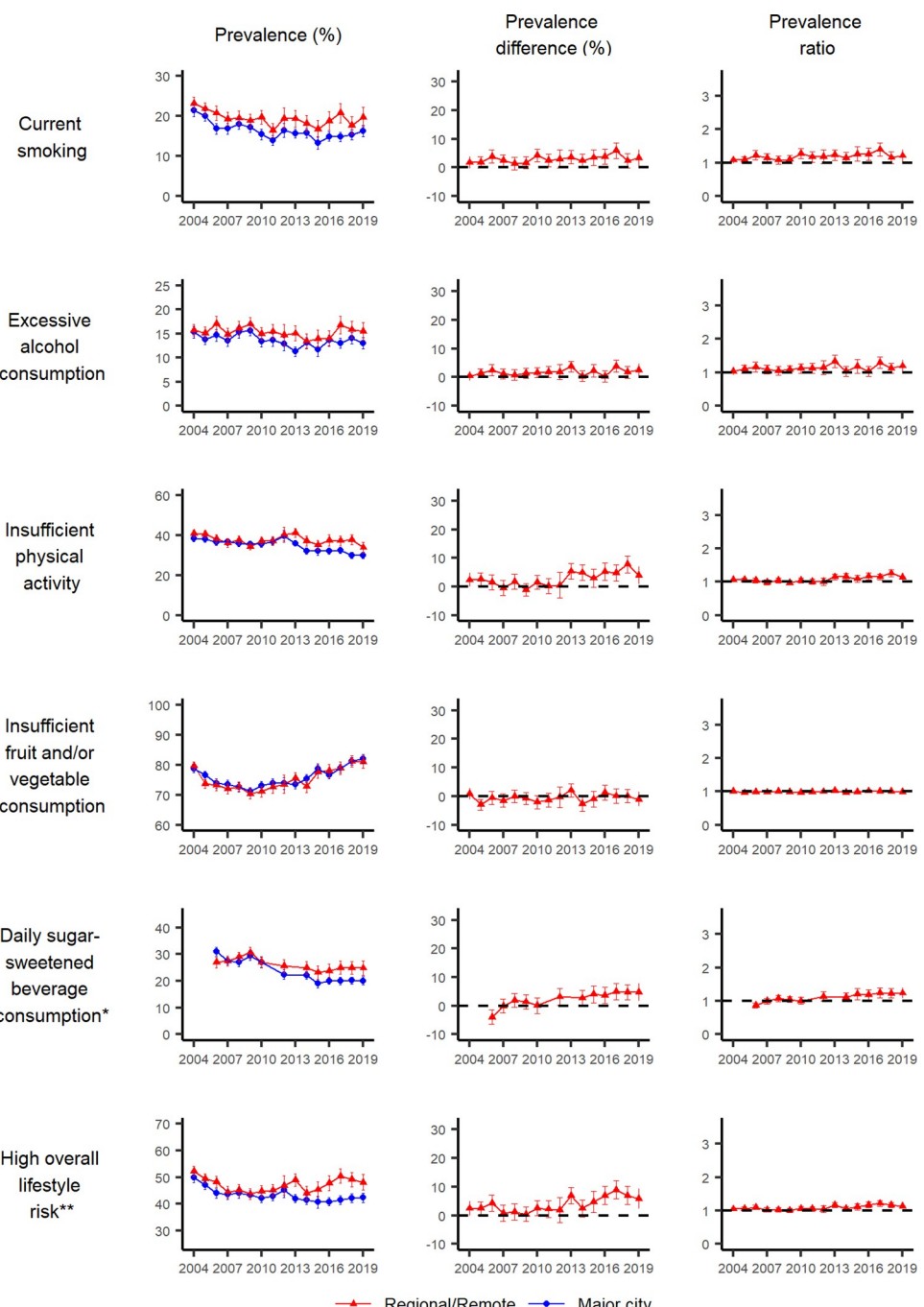

**Fig 4. Prevalence, prevalence difference and prevalence ratio (relative to major city), with 95% confidence intervals, of individual lifestyle behaviours and combined lifestyle risk index for the most and least disadvantaged groups for remoteness (ARIA+), by year, NSW adults aged 16 years and over, 2004–2019.** *The survey question related to daily sugar-sweetened beverage consumption was not asked in 2004, 2005, 2011 and 2013. ** Based on excessive alcohol consumption, insufficient physical activity, insufficient fruit and/or vegetable consumption and current smoking.

there was no clear increasing or decreasing trend over time in absolute and relative inequalities in excessive alcohol consumption by education. There was a relatively stable inequality by social disadvantage, and no substantial inequality by remoteness.

For physical activity, despite the suggested decline in insufficient physical activity overall, the prevalence of insufficient physical activity remained higher for the most disadvantaged subgroup compared to the middle and the least disadvantaged subgroups, based on education and area-level disadvantage, and to a lesser degree based on remoteness in more recent years. There appeared to be an increasing trend in absolute and relative inequalities between socioeconomic groups based on education, area-level disadvantage and remoteness. For example, the prevalence difference between the least and most disadvantaged education subgroups increased from 9.8 percentage points (95% CI: 6.3–13.2) in 2004 to 16.4 percentage points (95% CI: 12.2–20.5) in 2019, and that between those living in the least and most disadvantaged areas increased from 8.1 percentage points (95% CI: 3.8–12.4) to 12.4 percentage points (95% CI: 7.8–16.9). Similarly, the difference between those living in major cities and regional/remote areas increased from 2.3 percentage points (95% CI: -0.1–4.7) to 4.0 percentage points (95% CI: 0.9–7.1).

For insufficient fruit and vegetable consumption, the prevalence increased in all subgroups based on education, area-level disadvantage and remoteness from 2009 onwards. Based on education and area-level disadvantage, the prevalence was higher in the most disadvantaged subgroup compared to the least disadvantaged subgroups, but there was no visible difference by remoteness. There was no clear increasing or decreasing trend over time in absolute and relative inequalities by all three socioeconomic indicators.

For daily SSB consumption, the prevalence decreased in the middle and the highest education subgroups and remained similar over time for the lowest education subgroup. There were both increasing trends over time in absolute and relative inequalities by education and area-level disadvantage most notably from 2010 onwards, but to a lesser degree for remoteness. For example, the prevalence difference increased between the highest and lowest education subgroups from 10.3 percentage points (95% CI: 6.4–14.2) in 2010 to 19.1 percentage points (95% CI: 15.0–23.2) in 2019, while the prevalence ratio increased from 1.5 (95% CI: 1.3–1.7) to 2.5 (95% CI: 2.0–2.9). Similarly, the prevalence difference increased between the least and most disadvantaged subgroups from 10.1 percentage points (95% CI: 5.4–14.8) in 2010 to 17.7 percentage points (95% CI: 13.4–21.9) in 2019, and the prevalence ratio increased from 1.5 (95% CI: 1.2–1.7) in 2010 to 2.4 (95% CI: 1.9–2.9) in 2019.

## Combined lifestyle risk index

For all three socioeconomic indicators, the prevalence of an overall high-risk lifestyle (having two or more of the above risk factors) remained relatively stable in the most disadvantaged subgroup over time, despite visible improvements in other subgroups. There appeared to be a trend for worsening inequalities, indicated by both prevalence difference and prevalence ratios, primarily by education and area-level deprivation, and by remoteness to a lesser degree. The prevalence difference increased between the highest and lowest education subgroups from 11.7 percentage points (95% CI: 8.1–15.4) in 2004 to 22.0 percentage points (95% CI: 17.7–26.3) in 2019, while the prevalence ratio increased from 1.3 (95% CI: 1.2–1.4) to 1.6 (95% CI: 1.5–1.8). Similarly, the prevalence difference between the least and most disadvantaged subgroups increased from 7.8 percentage points (95% CI: 3.2–12.3) in 2004 to 17.1 percentage points (95% 12.1–22.1) in 2019, while the prevalence ratio increased from 1.2 (95% CI: 1.1–1.3) to 1.5 (95% CI: 1.3–1.6).

In the supplementary analyses in which daily SSB consumption was added to the lifestyle risk index (S2–S5 Tables), findings were similar overall to findings relating to the lifestyle risk index based on four individual behaviours.

## Discussion

Regular monitoring of multiple lifestyle risk behaviours and social inequalities over time is important for public health surveillance, policy planning and developing effective public health interventions to prevent chronic disease, especially among socially disadvantaged groups. In our study examining trends in multiple lifestyle risk behaviours from 2004 to 2019, we observed increasing socioeconomic inequalities in nearly all risk factors we examined, namely smoking, insufficient physical activity, daily SSB consumption, and the combined lifestyle risk index.

Overall, there were improvements between 2004 and 2019 in most health behaviours at the population level in NSW, with encouraging trends in smoking, physical activity, alcohol and SSB consumption. The prevalence of adults engaging in two or more lifestyle risk behaviours also decreased over time. These trends reflected those at the national level. The gradual reduction in smoking prevalence, largely reflecting current tobacco smoking prevalence as the overall prevalence of current e-cigarettes use was very low (about 1–2%), is consistent with the national trend which has seen tobacco smoking rates fall steadily over three decades [38, 39]. The prevalence of insufficient physical activity has also decreased, albeit slightly, in Australia in the last two decades [40]. Alcohol consumption has declined significantly over the last few decades in the Australian population overall [41], particularly in the younger adult population and among relatively light drinkers in the past decade [42]. There has been a decline between 1997 and 2018 in per capita volume sales of SSBs in Australia, mostly due to a fall in volume sales of sugar-sweetened carbonated soft drinks, suggesting a decline in the sugar contribution of SSBs the Australian diet [43]. In contrast, the prevalence for fruit and vegetable consumption appears to have stayed the same in the last 15 years at the national level [44] which differed from the fluctuating prevalence observed in this study.

Multiple factors may have contributed to the overall improvement in the healthy lifestyle profile of the population. These include better general awareness and knowledge of the importance of healthy living in the population over time. In NSW, several effective program and policies have also been implemented in the last decade to support healthy lifestyles [45–48]. Effective public health initiatives have included state-wide healthy eating and active living support programs such as the NSW Get Healthy Information and Coaching Service [48–50], the introduction of lockout legislation to curb alcohol-related violence in Sydney in 2014 which drew significant state-wide media attention, the introduction of plain packaging (Tobacco Plain Packaging Act and Regulations 2011) and health warnings on tobacco products in Australia in 2012. These population-wide programs and policies may have contributed to the overall improvement in lifestyle behaviours of adults living in NSW [29].

The prevalence of sufficient fruit or vegetable consumption in NSW, however, remained low during the study period. This may be due to many factors including the cost of fruit and vegetables relative to household income [51, 52], food preferences and poor recognition of inadequate fruit or vegetable consumption [53]. The prevalence of adults meeting recommended daily vegetable consumption has been notably lower than that of adults meeting recommended daily fruit consumption in NSW in the last two decades [54]. A state-wide and potentially national approach to promote fruit and vegetable consumption separately, and further research on effective interventions that increase vegetable consumption, including in more socially disadvantaged groups, may be needed [55].

A concerning finding from this study is that the most disadvantaged subgroups are less likely to have improved compared to the most advantaged. Although the prevalence decreased overall for most unhealthy lifestyle behaviours and the combined lifestyle risk index, it was still much higher in the most disadvantaged groups, especially based on education and area-level

disadvantage. State-wide programs such as free, telephone-based the Get Healthy Information and Coaching Service have been set up to enhance access and availability for all segments of the population, and this service has been shown to reach subgroups most in need including in socially and geographically disadvantaged regions [56, 57]. Access and availability for this type of service can, however, remain challenging for the most disadvantaged. Future policies and strategies in NSW may help to further address this issue by providing greater emphasis on equity focus and more substantial support (e.g. food price policy with price reductions for healthy items, interventions on the social and built environment). In contrast to the findings for other lifestyle behaviours, there was a tendency for the least disadvantaged subgroup based on area-level disadvantage to have a higher prevalence of excessive alcohol consumption than the most disadvantaged subgroups. This finding is broadly consistent with previous research conducted in Australia [58]. Different trends have also been observed among the drinking population [59], suggesting differences in abstention, and patterns of drinking, among those who do consume alcohol.

Another finding of concern was that socioeconomic inequalities appear to be widening for tobacco smoking, insufficient physical activity, daily SSB consumption, and the combined life-style risk indicator, based on one or several socioeconomic status indicators. This finding, which aligns with findings from several previous studies in Western countries [11, 15, 16], extends those from a previous study that also reported a widening gap for the 2002–2012 period in the NSW population, particularly in relation to smoking, fruit and vegetable consumption, and the combined lifestyle risk index [10]. The trade-off between population-level improvements versus widening socioeconomic inequalities aligns with the "inverse care law" first proposed by Julian Tudor Hart whereby those in most need of good health care are more likely to receive the least [13]. Effective ways to achieve better equity and address the widening gap in socioeconomic inequalities in health are needed to help people in lower socio-economic groups to improve their health behaviours. Proportionate universalism is one emerging approach that could be considered by government and policy makers [60, 61]. It relies on the concept that public health actions should be universal but with a scale and intensity proportionate to the level of disadvantage within the population. A framework for proportionate universalism has been proposed previously in the literature [62, 63], and trialled in various settings [64, 65], with responsibilities described for different levels of governance and the importance of creating partnerships [62].

## Strengths and limitations

Some of the strengths of the study were the relatively long study period, inclusion of individual and combined lifestyle risk factors, different individual- and area-level socioeconomic indicators, and consideration of both absolute and relative socioeconomic inequalities. The application of post-stratification weights helped to ensure population representativeness of the data. One limitation of the study was reliance on self-reported data which may be subject to recall and social desirability bias. Dietary risk could have been captured more comprehensively by including additional dietary variables. The data for daily SSB consumption was not available for all survey years.

## Conclusions

During the 2004 to 2019 period, there have been encouraging population-level improvements in most health behaviours including smoking, physical activity, alcohol and sugar sweetened beverage consumption, and the overall combined lifestyle risk. Conversely, those in the lowest socioeconomic groups were less likely to have improved compared to the most advantaged,

and the gap in socioeconomic inequalities has increased over time for the combined lifestyle risk index and several lifestyle behaviours, especially smoking, physical activity and daily SSB consumption. There is a clear need for governments to engage in ongoing risk factor surveillance and to implement or adapt strategies that seek to improve healthy lifestyle behaviours of the most disadvantaged populations.

## Supporting information

**S1 File. Questions on lifestyle behaviours from the 2019 NSW Population Health Survey Questionnaire.**
(DOCX)

**S2 File. Missing data.**
(DOCX)

**S1 Fig. Prevalence of being a current tobacco smoker and prevalence of current e-cigarette use[*] by year in NSW adults aged 16 years and over, 2004–2019.**
(DOCX)

**S2 Fig. Prevalence, prevalence difference and prevalence ratio of the combined lifestyle risk index[*] for the most and least disadvantaged groups for the three socioeconomic indicators, by year, NSW adults aged 16 years and over, 2006–2019.**
(DOCX)

**S1 Table. Patterns of missing data.**
(DOCX)

**S2 Table. Prevalence (with 95% confidence intervals) of individual lifestyle risk factors and combined lifestyle risk index, by year, persons 16 years and over, 2004–2019, NSW, Australia.**
(DOCX)

**S3 Table. Prevalence, prevalence differences and prevalence ratios of individual lifestyle risk factors and combined lifestyle risk index by educational attainment, by year, persons 16 years and over, 2004–2019, NSW, Australia.**
(DOCX)

**S4 Table. Prevalence, prevalence differences and prevalence ratios of individual lifestyle risk factors and combined lifestyle risk index by area-level disadvantage (IRSD), by year, persons 16 years and over, 2004–2019, NSW, Australia.**
(DOCX)

**S5 Table. Prevalence, prevalence differences and prevalence ratios of individual lifestyle risk factors and combined lifestyle risk index by geographical remoteness (ARIA+), by year, persons 16 years and over, 2004–2019, NSW, Australia.**
(DOCX)

## Acknowledgments

We acknowledge the NSW Ministry of Health who provided data for this study.

The work was completed while Janette Smith was employed as a trainee on the NSW Biostatistics Training Program funded by the NSW Ministry of Health. She undertook this work while based at the Prevention Research Collaboration, University of Sydney, Australia.

## Author Contributions

**Conceptualization:** Binh Nguyen, Philip Clare, Ding Ding.

**Formal analysis:** Janette Smith, Philip Clare.

**Methodology:** Binh Nguyen, Janette Smith, Philip Clare, Ding Ding.

**Supervision:** Binh Nguyen, Philip Clare, Ding Ding.

**Writing – original draft:** Binh Nguyen.

**Writing – review & editing:** Binh Nguyen, Janette Smith, Philip Clare, Philayrath Phongsavan, Leonie Cranney, Ding Ding.

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
