## [Decision Letter · Decision Letter 0]

21 Dec 2022

PGPH-D-22-01527

Sixteen-year trends in multiple lifestyle risk behaviours by socioeconomic status from 2004 to 2019 in New South Wales, Australia

Dear Dr. Nguyen,

Thank you for submitting your manuscript to PLOS Global Public Health. After careful consideration, we feel that it has merit but does not fully meet PLOS Global Public Health’s publication criteria as it currently stands. Therefore, we invite you to submit a revised version of the manuscript that addresses the points raised during the review process.

Both reviewers provided favorable recommendations and suggested minor changes in the manuscript. Thank you for submitting such a well-written manuscript and please address the issues raised by the reviewers in your revision.

We look forward to receiving your revised manuscript.

Kind regards,

Biplab Datta, Ph.D.

Academic Editor

Journal Requirements:

1. We do not publish any copyright or trademark symbols that usually accompany proprietary names, eg (R), (C), or TM  (e.g. next to drug or reagent names). Please remove all instances of trademark/copyright symbols throughout the text, including R on page 20.

Additional Editor Comments (if provided):

Reviewers' comments:

Reviewer's Responses to Questions

**Comments to the Author**

1. Does this manuscript meet PLOS Global Public Health’s publication criteria? Is the manuscript technically sound, and do the data support the conclusions? The manuscript must describe methodologically and ethically rigorous research with conclusions that are appropriately drawn based on the data presented.

Reviewer #1: Yes

Reviewer #2: Yes

2. Has the statistical analysis been performed appropriately and rigorously?

Reviewer #1: Yes

Reviewer #2: Yes

3. Have the authors made all data underlying the findings in their manuscript fully available (please refer to the Data Availability Statement at the start of the manuscript PDF file)?

Reviewer #1: Yes

Reviewer #2: Yes

4. Is the manuscript presented in an intelligible fashion and written in standard English?

Reviewer #1: Yes

Reviewer #2: Yes

5. Review Comments to the Author

Reviewer #1: I enjoyed reading this well-written, important paper. The methodology was well prepared, the results well described and argued in the discussion. The conclusions are relevant to the field of public health. The entire paper is supported with the relevant literature. I wish authors all the best and look forward to seeing the paper published.

Reviewer #2: Comments to author

The authors sophisticatedly and carefully examine long-term changes in the prevalence of various behavioral risks and the socioeconomic disparities therein. The methods used are appropriate, and the authors’ findings are inspiring. This is a very well-written manuscript. However, I have some questions about modeling decisions and interpretations that, if clarified, would strengthen the paper. These, along with more minor questions/suggestions, are listed below.

* Could the authors provide more justifications for why ARIA+ can be a measure of socioeconomic status? For example, although there are fewer healthcare services in rural areas, whether one is disadvantaged depends on his/her wealth. That is, wealthy people have cars and may buy houses in rural areas, whereas poor people may live in the cities, but poverty may restrict their access to health-related resources.

* Could the authors elaborate why area-level indicators are better than individual-level indicators for capturing respondents’ socioeconomic status? Could the authors explain why SEIFA is a better measure than individual-level wealth/income?

* P.7-8: Could the authors elaborate why controlling for both immigration status and linguistic backgrounds is necessary? I am wondering why “whether they spoke a language other than English at home” matters, since it seems not necessarily related to respondents’ English proficiency.

* P.8: Is the distribution of the lifestyle risk index right-skewed? Would using poisson or negative binominal regression change the results?

* P.15: The authors said, “For current smoking, the prevalence decreased for all education subgroups over time”. However, in Figure 2, CIs for the smoking prevalence in the least educated group largely overlap across years. This is also the case for insufficient physical activity and lifestyle risk index.

* P16: It seems that the authors did not interpret results regarding the remoteness differences in excessive alcohol consumption.

* P20: It seems to me that the prevalence of insufficient fruit/vegetable consumption tends to increase over time, which is more evident in Figure 3-5. Could the authors add more discussions on what potential reasons would be even if there is a state-wide healthy eating program?

* P21: Regarding the “counterintuitive” results of excessive alcohol assumption, it is documented that the relationship between drinking behavior and health outcomes depends on the quality and types of alcohol. Perhaps the authors can interpret observed patterns in this way.

* Figure 1: Could the authors add a line for the means of prevalence so that readers can easily identify whether the prevalence increases or decreases?

6. PLOS authors have the option to publish the peer review history of their article (what does this mean?). If published, this will include your full peer review and any attached files.

**Do you want your identity to be public for this peer review?** For information about this choice, including consent withdrawal, please see our Privacy Policy.

Reviewer #1: No

Reviewer #2: No

---

## [Editor Report · Decision Letter 1]

25 Jan 2023

Sixteen-year trends in multiple lifestyle risk behaviours by socioeconomic status from 2004 to 2019 in New South Wales, Australia

PGPH-D-22-01527R1

Dear Ms Nguyen,

We are pleased to inform you that your manuscript 'Sixteen-year trends in multiple lifestyle risk behaviours by socioeconomic status from 2004 to 2019 in New South Wales, Australia' has been provisionally accepted for publication in PLOS Global Public Health.

Best regards,

Biplab Datta, Ph.D.

Academic Editor